biomathematics/computational biology/mathematical modelling

influenza virus, immune system, repeated influenza vaccinations, repeated influenza infections, mathematical modelling

**Author for correspondence:**
Emmanuel S. Adabor
e-mail: emmanuelsadabor@gimpa.edu.gh

†Present address: School of Technology, Ghana Institute of Management and Public Administration, PO Box AH50, Achimota, Accra, Ghana.

# Computational investigations of the immune response to repeated influenza infections and vaccinations

Emmanuel S. Adabor[1,2,†]

[1]Reserch Centre, African Institute for Mathematical Sciences, Cape Town, South Africa
[2]Department of Mathematical Sciences, Stellenbosch University, Stellenbosch, South Africa

ESA, 0000-0003-4471-1915

Previous studies have shown that repeated influenza vaccination can enhance susceptibility to subsequent infection with a drifted influenza virus strain. This paper seeks to further understanding of the interactions between influenza viruses and specific immune cells that accompany this phenomenon. The paper argues that repeated vaccination increases susceptibility to infection only in the context of a residual immunity induced by prior vaccination or infection. The results of computational analysis indicate that this is a dynamic consequence of interactions between vaccines, influenza viruses and specific immune cells. In particular, mathematical modelling was used to show that in the presence of residual immunity conferred by a vaccine administered in Canada in the 2013–2014 influenza season, the 2014–2015 season vaccine enhanced susceptibility to infection. Such infection enhancement occurs when the 2014–2015 vaccine boosts suppressive T-regulatory cells induced by the 2013–2014 vaccine, decreasing the strength of antibody responses to the infecting strain. Overall, the study suggests probable characteristics of infecting viruses and vaccines that make repeated influenza infections and vaccinations detrimental.

## 1. Introduction

The human immune response to infections is a complex system in which a series of mechanisms are instantiated to react to antigens. Previous exposures of the immune system to antigens facilitate prompt responses to future infections with the same antigens. The body accomplishes this by using immunological memory cells derived from the previous exposures to the antigens. Besides direct natural infections from disease causing organisms resulting in what is termed as natural immunity, vaccinations intended to stimulate specific immune responses are also means

to expose the body to antigens to confer resistance to future infections. The latter is an efficient medical intervention that has been successfully applied to eradicate mortality from some communicable diseases over the years [1]. However, for some reasons, there has been variable susceptibility to infections even with previous vaccinations to some disease causing microbes. It has been observed that rather than enhancing immunity, some previous infections and vaccinations are sometimes counter-protective. Several lines of studies have suggested the occurrence of this phenomenon in relation to influenza virus, dengue, human immune deficiency virus (HIV) and rabies [2–8].

Particularly, this has been well reported in the studies of influenza since 1979 when Hoskins and colleagues revealed that repeated influenza vaccinations only provided a suboptimal protection in the long term [9]. Recently, a study further suggested single vaccinations had higher vaccine effectiveness compared with repeated vaccinations in the 2014–2015 influenza season in Canada [4]. In that study where the 2013–2014 and 2014–2015 seasons' vaccines had identical influenza A (H3N2) components, vaccine effectiveness against influenza A (H3N2) for 2014–2015 influenza season was ascertained from participants who had received vaccines for two consecutive seasons (2013–2014 and 2014–2015) and for only one season (2013–2014 or 2014–2015) [4]. The study by Skowronski and colleagues realized that taking the 2013–2014 vaccine alone (single vaccination) conferred negligible immunity and that there was no evidence of infection enhancement [4]. In contrast, taking both the 2013–2014 and 2014–2015 seasons' vaccines (repeated vaccination) enhanced infection by the epidemic virus [4]. These results on influenza vaccine effectiveness against the influenza A (H3N2) during the 2014–2015 influenza season are consistent with other studies conducted elsewhere [10,11]. In order to investigate these scenarios, the study posits that repeated vaccination increases susceptibility to influenza infections only in the context of a residual immunity induced by vaccination or infection. This is analysed through a comparison between two groups, single and repeated vaccinations, with residual immunity being conferred by first vaccination.

Using a novel mathematical model, this study investigates the immunological dynamics underlying these particularly troubling observations in influenza. More specifically, separate systems of models designed to capture the immune responses under various conditions of repeated vaccinations and infections are conveyed in a system of differential equations. In this way, it will be possible to ascertain the consequences of synergies of influenza viruses, specific immune cells and vaccines to describe the immune response in both repeated infections and vaccinations. With an elucidated immune response under these conditions, this paper further provides possible implications of findings on the effectiveness of influenza vaccines.

# 2. Material and methods

## 2.1. Mathematical model

The model reproduces the consequences of the dynamic interactions between the immune agents and vaccination during influenza epidemic. It comprises of a subset of the components of the immune system, the regulation of immune response to influenza virus, T-regulatory cells (tregs) and the target cells of the virus.

The influenza virus infects epithelial cells of the lungs which results in the activation of lymphocytes through mechanisms involving antigen presentation by antigen presenting cells [12,13]. When an influenza virus infects its target cells, the immune system is stimulated to produce antibodies to promote clearance of the virus from the body. In this process, virus antigens are engulfed by antigen presenting cells such as dendritic cells. Antigen particles are loaded on the surfaces of the dendritic cells attached to MHC II (major histocompatibility complex). The antigen-MHC II complex leads to the activation of T cells and then the proliferation and differentiation of B cells [14,15]. B cells produce finely tuned antibodies that bind to the specific virus particle and mark them for destruction. However, some B cells are changed into immune memory cells to facilitate a swift immune response for future infection with the same antigen. While the immune system undergoes the process to react to the antigen, the virus particle replicates and in some cases may have undergone mutations [16,17]. As a result, a future infection by an antigenic variant of the virus leaves the immune system trapped by the response to the first viral particle and so efficient reaction is hampered. This phenomenon motivates the use of models in which viruses are antigenically distinct to investigate the immune response to repeated influenza infections/vaccinations in this study.

Particularly, in line with previous studies [18–20], this study focuses on humoral aspects of adaptive immunity. For a strain V1, if the adaptive immune system has never encountered any antigen which cross-reacts with V1, specific naive lymphocytes will be activated [21,22]. When a strain V2, which cross-reacts with V1, is encountered by the immune system in future infection, then a subset of V1-specific lymphocytes, including tregs will be re-activated in addition to the activation of naive lymphocytes by V2. The reactivated tregs will suppress the presentation of V2-specific antigens by dendritic cells and decrease antigen dose loading of V2 accessible for activating V2-specific B cells and specific T-helper cells [22,23]. If a strain V3 that cross-reacts with V1 and V2 is encountered, then the strength of humoral response to V3 will depend on activated V1- and V2-specific B cells and naive lymphocytes elicited by V3 [23]. Note that, a much lower antigen dose will be needed to reactivate the V1- and V2-specific lymphocytes compared with naive lymphocytes [24]. These facts about humoral aspects of adaptive immunity form the structure of the model. In particular, it covers interactions between the virus, its target cells and the regulation of the humoral response by tregs. Therefore, the equations of the simplified model (equations (2.1)–(2.9)) were formulated based on these interactions. This approach is consistent with other prior studies and sufficient in details regarding the scope of this study [18–20]. In the model, cross-reactive rates specify cross-reactivity between strains. Here, two strains cross-react when lymphocytes activated by one strain can recognize the other strain. Further details about the model simulation of sequential infection and vaccination are presented in §2.2. The variables and parameters of this model are described in table 1. A simple overview of the dynamics of the aspects of immune response considered in the modelling is summarized in figure 1.

With the background of these aspects of the adaptive immunity which has also been the main focus of previous studies [18–20], the model depicts the interactions between the virus, the infected cells, the humoral immune response and regulation by tregs with a focus on effector B cells and antibodies (Abs). This approach of presenting such dynamics is consistent with the literature [27]. More specifically, influenza virus of strain $i$, infects target cells ($E_i$) at a rate $\beta_E V_i$ leading to the activation of B cells ($B_i$) and tregs ($R_i$). The activated virus-specific B cells also produce IgG Abs (with concentration $A_i$ pg ml$^{-1}$) that react to the virus. As has been indicated, previous knowledge [15] permits a relationship to be assumed between the activation of B cells and antigen dose loaded ($D$). However, antigen presentation is suppressed by activated tregs [23] which is marked by the relationship between the total number of activated tregs ($R$) and antigen dose loaded ($D$). These motivated equations (2.3) and (2.4) in the model. Equations (2.1) and (2.2) mark the dynamics of the number of healthy target cells and the number of infected target cells respectively over time. The presentation of virus antigens by dendritic cells results in the activation of B cells, which produce virus-specific antibodies. This dynamic relationship informed the formulations of equations (2.5) and (2.6) of the model. In addition, the infecting virus titre (with the total concentration $V$) at any time will be expected to be a function of the number of infected cells and the concentration of the neutralizing Abs as entailed in equation (2.7). The mathematical formulations based on these facts founded the following model:

$$\frac{d\bar{E}}{dt} = p_E(E_0 - \bar{E}) - \beta_E \bar{E}V, \tag{2.1}$$

$$\frac{dE_i}{dt} = \beta_E \bar{E}V_i - \delta_E E_i \tag{2.2}$$

$$\frac{dD_i}{dt} = \beta_D(1 - D_i)V_i - K_R D_i R, \tag{2.3}$$

$$\frac{dR_i}{dt} = b_R[1 - f(D_i, \tau_n, q)]f(D_i, \tau_n, q) + p_R R_i f\left(\sum_j \sigma_{ij}D_j, \tau_a, q\right) - \delta_R R_i, \tag{2.4}$$

$$\frac{dB_i}{dt} = b_B f(D_i, \eta_n, q) + p_B B_i f\left(\sum_j \sigma_{ij}D_j, \eta_a, q\right) - \delta_B B_i, \tag{2.5}$$

$$\frac{dA_i}{dt} = \varepsilon_A B_i - \delta A_i, \tag{2.6}$$

$$\frac{dV_i}{dt} = \varepsilon_V E_i - \left(c_V + c_A f\left(\sum_j \sigma_{ij}A_j, \lambda, s\right)\right)V_i, \tag{2.7}$$

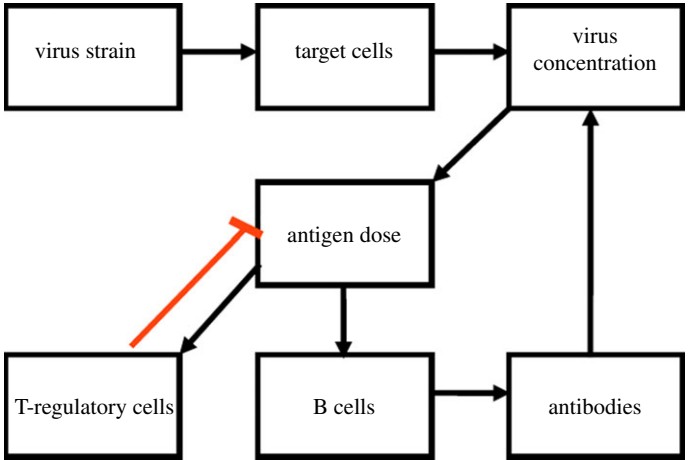

**Figure 1.** A simple model of the dynamic interactions between immune cells and antigens. The influenza virus infects the target cell and replicates. Dendritic cells load antigen dose to facilitate the production of T cells and then B cells. T-regulatory cells dynamically control the antigen presentation by dendritic cells. B cells produce antibodies which mark the virus leading to their clearance from the body.

where

$$f(x, y, z) = \frac{x^z}{y^z + x^z} \tag{2.8}$$

is a saturation function and $i$ or $j$ identifies the specific virus strain.

An addition to this model is the extension to simulate influenza vaccination. Particularly, in vaccinations, there are no rates of non-specific virus clearance and infections of vaccine strains. These are particularly significant since immunizations are given wholly to individuals and the vaccine strains do not replicate. Therefore, a new equation accounting for these variations in equation (2.7) excludes rate of non-specific virus clearance ($c_v$) and this is given by:

$$\frac{\mathrm{d}V_i}{\mathrm{d}t} = \varepsilon_V E_i - \left( c_A f \left( \sum_j \sigma_{ij} A_j, \lambda, s \right) \right) V_i. \tag{2.9}$$

Furthermore, in order to support the results of this study, the model was tested on its ability to predict *in vivo* dynamics of influenza virus infection and IgG antibodies concentration measured in the lungs and sera of infected mice reported in previous studies [25]. The model provided a reasonable fit to the experimental data (figure 2) published in [25,27].

## 2.2. Design of experiments

Using the mathematical model as a tool, the study simulates in different groups (vaccination and infections) the exposition of the immune system to specific influenza virus strains and then elicits its response to a cross-reactive virus in an epidemic. In the repeated infections, the case of three cross-reactive virus strains, namely $V1$, $V2$ and $e$ will be considered. In the adaptive immune system, the first infection with the virus strain $V1$ will activate naive lymphocytes specific to react to $V1$. Therefore, $V1$-specific immune cells will be readily activated to react to a subsequent $V1$ infection. If a virus strain, $V2$ which cross-reacts with $V1$ is encountered, a subset of the $V1$-specific lymphocytes will be reactivated to react to $V2$ depending on the rate of cross-reactivity between the two strains. Besides, a subsequent infection in an epidemic with the virus strain, $e$, which cross-reacts with both $V2$ and $V1$ will reactivate naive immune cells as well as immune cells with specificities for $V2$ and $V1$. Using a sequential infection with three virus strains, the model can be readily reproduced to elicit the immune response in terms of the variables in the model (table 1). Consequently, the subscripts $i$ and $j$ in the model take the values 1, 2 and 3 to denote the infecting strain being considered. Hence, the model is resolved into a system of differential equations of the immune components derived from these sequential infections. Similarly, sequential vaccinations are simulated and followed by a circulating virus infection. The design of the experiments is consistent with previous research that identified the last two vaccinations as the most important determinants of the efficacy of vaccines in an epidemic [29].

**Table 1.** Variables and parameters of the study.

| variables/ parameters | definition (units) | value [ref.] |
|---|---|---|
| $V$ | infectious virus titre (EID$_{50}$ ml$^{-1}$) | determined |
| $\bar{E}$ | number of uninfected target cells | determined |
| $E$ | number of infected target cells | determined |
| $E_0/\bar{E}(0)$ | initial number of uninfected target cells | $2 \times 10^5$ [25] |
| $R$ | number of activated T-regulatory cells (tregs) | determined |
| $B$ | number of activated B cells | determined |
| $A$ | IgG titre in serum (pg ml$^{-1}$) | determined |
| $D$ | antigen dose loaded by dendritic cells | determined |
| $p_E$ | renewal rate of target cells (day$^{-1}$) | $1 \times 10^{-3}$ [25] |
| $\beta_E$ | infection rate of target cells (ml EID$_{50}$$^{-1}$ day$^{-1}$) | $5 \times 10^{-6}$ [25] |
| $c_V$ | rate of non-specific virus clearance (EID$_{50}$ ml$^{-1}$ day$^{-1}$) | 4 [25] |
| $\varepsilon_v$ | production rate of infectious virus per infected target cell (EID$_{50}$ ml$^{-1}$ day$^{-1}$) | $1 \times 10^2$ [25] |
| $b_B$ | maximum activation rate of naive B cells (day$^{-1}$) | 3 [26] |
| $b_R$ | maximum activation rate of naive tregs (day$^{-1}$) | 3 [26] |
| $p_B$ | maximum proliferation rate of B cells (day$^{-1}$) | $8 \times 10^{-1}$ [27] |
| $p_R$ | maximum proliferation rate of tregs (day$^{-1}$) | $8 \times 10^{-1}$ [27] |
| $\varepsilon_A$ | production rate of IgG antibody by B cells | $6 \times 10^{-2}$ [26] |
| $\delta_B$ | death rate of B cells (day$^{-1}$) | $1 \times 10^{-1}$ [26] |
| $\delta_R$ | death rate of tregs (day$^{-1}$) | $1 \times 10^{-1}$ [26] |
| $\delta_E$ | death rate of infected target cells (day$^{-1}$) | 1.2 [25] |
| $\delta_A$ | clearance rate of IgG antibody (day$^{-1}$) | $4 \times 10^{-2}$ [26] |
| $S$ | average number of antibodies for virus neutralization | 3 [28] |
| $Q$ | average number of dendritic cell-loaded antigens for lymphocyte activation | 1 [27] |
| $c_A$ | rate constant for virus neutralization by IgG (ml pg$^{-1}$ day$^{-1}$) | $1 \times 10^2$ [27] |
| $\tau_n$ | antigen dose for half-maximal activation of naive tregs | $3 \times 10^{-1}$ [27] |
| $\tau_a$ | antigen dose for half-maximal reactivation of pre-activated tregs | $1 \times 10^{-2}$ [27] |
| $\eta_n$ | antigen dose for half-maximal activation of naive B cells | $3 \times 10^{-1}$ [27] |
| $\eta_a$ | antigen dose for half-maximal reactivation of pre-activated B cells | $1 \times 10^{-2}$ [27] |
| $\lambda$ | antibody concentration for half-maximal neutralization of virus (pg ml$^{-1}$) | $1 \times 10^3$ [27] |
| $\beta_D$ | rate of antigen loading by dendritic cell (ml EID$_{50}$$^{-1}$ day$^{-1}$) | $1 \times 10^{-3}$ [27] |
| $K_R$ | rate of antigen de-loading by dendritic cell under the influence of tregs (day$^{-1}$) | $1 \times 10^{-1}$ [27] |
| $\sigma_{ij}$ | cross-reactive rate of strain $i$ to strain $j$ | ranges between 0 and 1 |

In the absence of any infections, there will be no infected cell, virus-specific B cells, tregs, antigen dose to be loaded by dendritic cells, and virus-specific antibodies at the beginning of the experiment. This is indicated by the initial conditions of the variables in the system. However, the initial number of uninfected target cells and the available concentration of infecting virus at the start of the experiments were set to 200 000 and 1500 EID$_{50}$ ml$^{-1}$, respectively [25,27]. The rest of the values for simulating infections are presented in table 1. The systems simulating both repeated infections and vaccinations are modelled and solved in R with the aid of deSolve package [30].

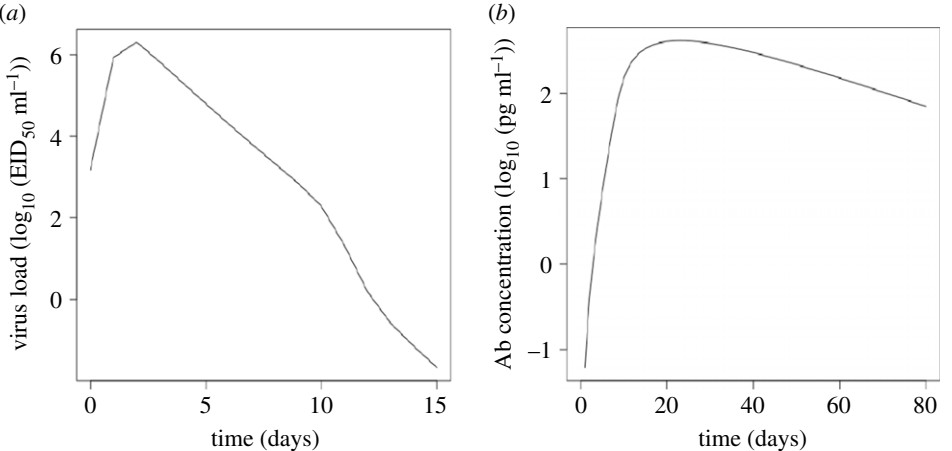

**Figure 2.** Model fit experimental data. (*a*) Model fit to viral load measured in the lungs of infected mouse reported in [26]. (*b*) Model fit to IgG antibody concentration measured in the sera of infected mouse [26].

# 3. Results

## 3.1. Repeated sequential vaccinations and infections

To examine the effects of repeated vaccinations on the immune cells, experiments in two groups were simulated using the proposed model. In the first group (repeated vaccination), the first vaccination was performed with the vaccine strain derived from the virus $V1$ for 28 days. This was followed by another vaccination with vaccines derived from the strain $V2$ for another 28 days. The second group (single vaccination) was vaccinated with only one vaccine derived from strain $V1$ for 28 days. Each group and a naive group without any prior vaccination are then infected with equal amounts of virus titre of the challenge strain, $e$ (epidemic strain), that cross-reacts with each vaccine strain. The peak viral load in each group is derived for different rates of cross-reactivity between vaccine strains (vaccine–vaccine cross-reactivity) and between vaccine strains and epidemic strain (vaccine–epidemic strain cross-reactivity). Although 10% and 20% rates of vaccine–vaccine cross-reactivity are considered, the vaccine–epidemic strain cross-reactivity was varied between 10% and 90%. On the other hand, to determine the effects of prior infections on immune response in an epidemic, simulations of sequential infections were performed in another two different groups with virus strains in a mathematical model.

In the first group (repeated infection), the sequential infection was simulated with strain $V1$ for 28 days followed by infection with strain $V2$ for another 28 days. In the second group (single infection), the model simulated the infection with $V1$ alone for 28 days. The simulation in this second group was allowed for another 28 days before the challenge infections were applied. Each group is then challenged with $e$ (epidemic strain) infection for 28 days. Here, while cross-reactive rates of 10% and 20% between infecting strains prior to the epidemic infections (prior strains cross-reactivity) were used, the cross-reactivity between epidemic and previous infecting strains (prior–epidemic strain cross-reactivity) was varied from 10% to 90%. In each experiment of this study, the immune response in terms of the activated effective tregs, effective B cells and Abs as well as the peak viral loads are derived during the challenge infection. The peak virus load in each experiment was calculated as the peak amount of the challenge infection minus the challenge dose [27].

## 3.2. Viral loads

For a vaccine–vaccine cross-reactivity of 10%, viral loads in repeated vaccination were reducing for vaccine–epidemic strain cross-reactivity up to 70% as did single vaccination (figure 3). However, the results revealed significant differences in viral loads between single and repeated vaccinations for lower vaccine–epidemic strain cross-reactivity. In each vaccinated group, viral loads diminished as the cross-reactivity between vaccine and the epidemic strains increased. This was to be expected since increasing vaccine–epidemic strain cross-reactivity increases the activated lymphocytes for clearing virus [27]. Nevertheless, in repeated vaccination, sufficient lymphocytes to completely annihilate virus is achieved when the vaccine–epidemic strain cross-reactivity is 50% while this is only possible at 70%

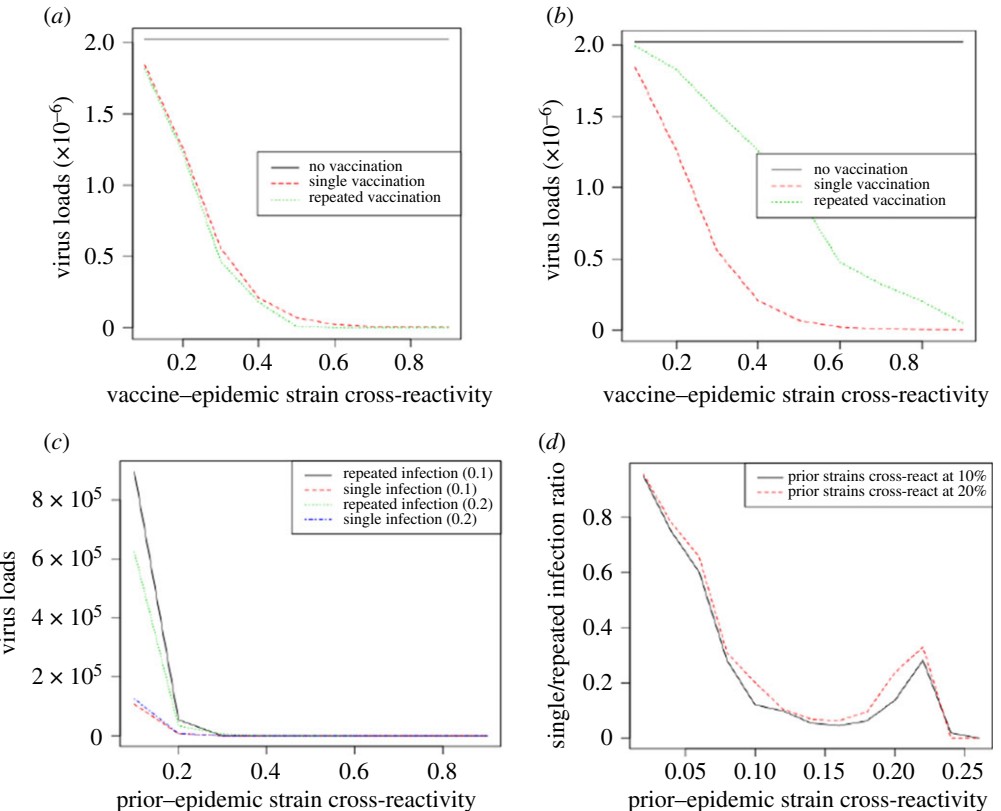

**Figure 3.** Dynamics of viral loads in repeated vaccination and infection. (*a*) Viral loads (EID$_{50}$ ml$^{-1}$) when vaccine strains cross-react at 10%. (*b*) Viral loads (EID$_{50}$ ml$^{-1}$) for vaccine–vaccine cross-reactivity of 20%. (*c*) Viral loads (EID$_{50}$ ml$^{-1}$) in repeated infections. (*d*) Ratios of viral loads in repeated infections. Viral loads corresponding to other rates of vaccine–vaccine cross-reactivity are presented in electronic supplementary material, figure S1.

in single vaccination (figure 3*a*). In the experiment with vaccine–vaccine cross-reactivity of 20%, higher viral loads were found over the period of increasing the vaccine–epidemic cross-reactivity in as much as it was falling in repeated vaccination (figure 3*b*). This was to be expected for different vaccine–vaccine cross-reactivity. The viral loads increased in the repeated vaccination as a result of lower effective lymphocytes activated to react to the epidemic permitting virus to replicate. Higher viral loads in repeated vaccination are indicative that they are more susceptible to the influenza epidemic infections compared with single vaccination. These observations were not different in repeated sequential infections as they had the highest viral loads compared with single infection although they were characterized by lower viral loads compared with repeated vaccination (figure 3*c*). However, although no viral load was found in each infection group beyond prior–epidemic strain cross-reactivity of 35%, it was found that the single infection had much lower values with no viral load from 20% in all experiments (figure 3*c*). When prior infections cross-reacted at 10%, no viral load was observed at prior–epidemic strain cross-reactivity of 30% or more. Nonetheless, the absence of viral loads could only be found after a prior–epidemic strain cross-reactivity of 35% in the case when prior infections cross-react at 20% (figure 3*c*). For the experiment in which prior infections cross-react at 20%, the viral loads at the early epidemic infection require an increased amount of activated lymphocytes which is achieved when the prior–epidemic strain cross-reactivity increases beyond 30%.

To further explore the viral loads, the study investigates their ratios within the range of cross-reactivity where significant viral loads were found in the repeated infection. The ratios represent the fraction of viral loads of single infection contained in repeated infection. Overall, the ratios decreased to less than 10% as the prior–epidemic strain cross-reactivity increased to 16% (figure 3*d*). This consistent decrease was anticipated since the viral loads in single and repeated infections decreased together (figure 3*c*). Increasing the prior–epidemic strains cross-reactivity enhances the activation of effective lymphocytes to react to the epidemic strain. This reduces the viral loads in each infection. However, as the prior–epidemic cross-reactivity exceeds 16%, the single infection has relatively very small and constant viral loads compared with the decrements observed in the repeated infection. This

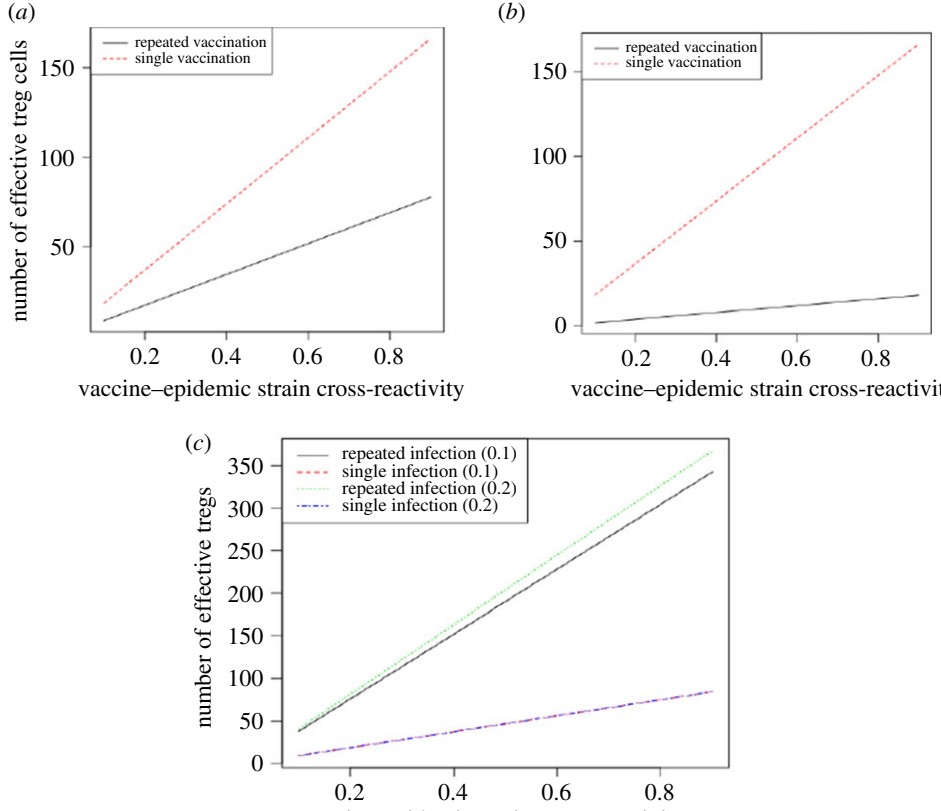

**Figure 4.** Effective T-regulatory cells activated by epidemic infection. (*a*) Number of effective tregs when vaccine strains cross-react at 10% in repeated vaccination. (*b*) Number of effective tregs when vaccine strains cross-react at 20% in repeated vaccination. (*c*) Number of effective tregs in repeated sequential infections.

increases the ratios above 20% until the prior–epidemic strain cross-reactivity is 22% (figure 3*d*). The higher viral loads in repeated infections in the experiment involving prior strains that cross-react at 20% accounted for higher ratios as the cross-reactive rate between the epidemic strain and the prior infections increases to 22%. These results suggest that the viral loads in single infection only exceed half of viral loads in repeated infection while the prior–epidemic strain cross-reactivity is not more than 6%. However, it is largely reduced when the cross-reactive rate between the epidemic strain and the prior infections exceeds 6%. Thus, repeated infection group becomes more susceptible to the epidemic infections because they have higher viral loads compared with single infections.

## 3.3. Dynamics of immunity under repeated vaccinations and infections during epidemics

During vaccinations, there are lower amounts of activated lymphocytes compared with prior infections [31]. These decreased amounts are incorporated in the effective lymphocytes activated to react to epidemic strains which cross-react with the vaccine strains. Therefore the amount of effective lymphocytes derived after vaccination in an epidemic is lower compared with effective lymphocytes derived from sequential infections. As a result, the observations in respect of effective tregs, effective B cells and effective Abs produced to react to the epidemic strain were in lower amounts in the vaccination experiments compared with the sequential infections. These account for higher viral loads identified in the vaccination compared with prior infection experiments. Consequently, low effective tregs were found in vaccination experiments compared with the sequential infections (figure 4).

Most importantly, the number of effective tregs in the repeated vaccinations was lower compared with single vaccination (figure 4). Vaccine strains that cross-react at 20% culminate in decreasing the antigen dose loaded by dendritic cells for activating lymphocytes specific to the vaccine strains in repeated vaccination. Therefore the effective tregs activated during the epidemic is lower compared with experiments in which vaccine strains cross-react at 10%. In addition, the effective tregs increased as the vaccine–epidemic strains increased in each vaccination experiment (figure 4). These results were obtained since increasing the vaccine–epidemic strain increases the reactivated tregs that enhance the effective tregs derived

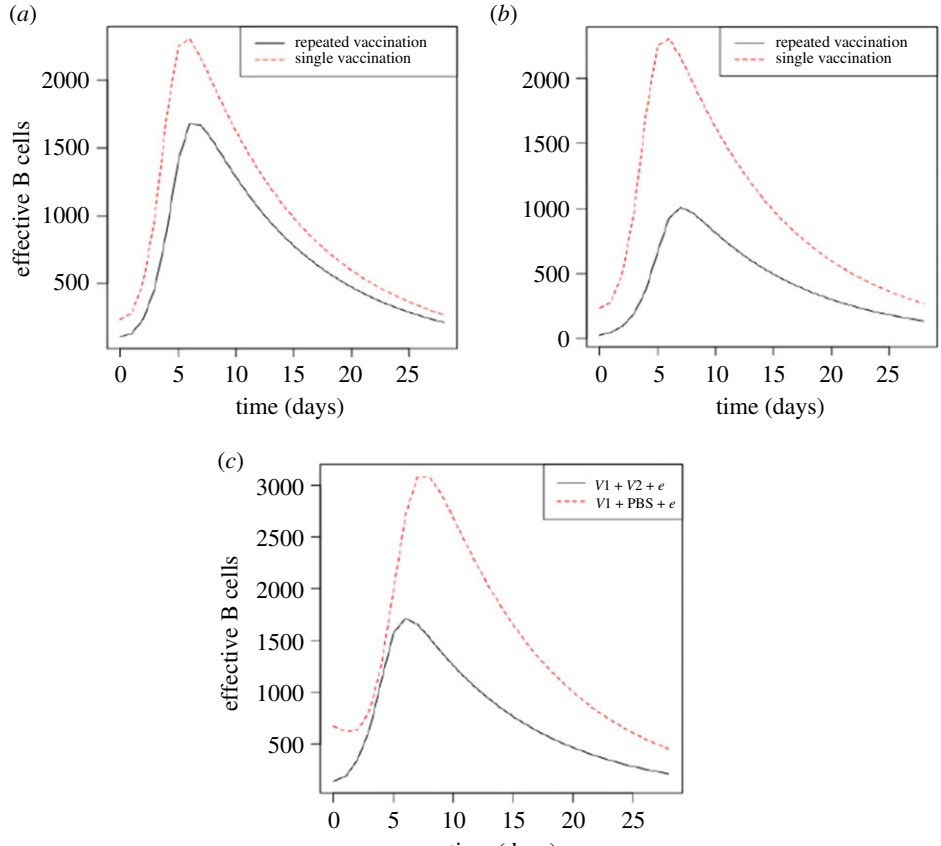

**Figure 5.** Effective B cells activated by epidemic infection. (*a*) Effective B cells activated in repeated vaccination when prior vaccine strains cross-react at 10%. (*b*) Effective B cells activated in repeated vaccination when vaccine strains cross-react at 20%. (*c*) Effective B cells activated in sequential infections. $V1 + V2 + e$ is repeated infection and $V1 + PBS + e$ is single infection.

during the epidemic infection. On the contrary, the effective tregs activated in repeated infections are extremely high compared with single infection during the epidemic (figure 4*c*). The higher numbers of activated effective tregs in the repeated infections are accumulated from reactivation of the tregs derived from the two prior infections that cross-react with the current epidemic strain. Nevertheless, the rising numbers of tregs in all experiments as prior–epidemic strain cross-reactivity increased resulted from increasing reactivated tregs for higher cross-reactive rates. Higher vaccine– or prior–epidemic cross-reactivity increases the reactivated tregs derived in reaction to the epidemic. As a result, the effective tregs increase during the epidemic as more tregs are activated in respect of epidemic strain. Unlike repeated vaccination, repeated infections in which the prior strains ($V2$ and $V1$) cross-react at 20% had higher tregs compared with single prior infection (figure 4). This was to be expected since the epidemic strains reactivate larger subsets of the cumulated tregs in those with 20% cross-reactivity compared with 10% cross-reactivity. Tregs are important moderators of antigen presentation by dendritic cells for the activation of B cells in repeated infections [23].

As 20% vaccine–vaccine cross-reactivity culminates in decreasing the fraction of sites on dendritic cells that load antigens, the activated effective B cells and the consequential antibodies (Abs) produced during the epidemic infection are lower compared with the 10% vaccine–vaccine cross-reactivity (figures 5 and 6). Furthermore, as epidemic strain is cleared, the amount of antigens available for loading by dendritic cells also decreases leading to a decrease in the activated effective B cells after day 7 in each experiment (figure 5). However, the presence of virus replicating at the early days of infection stimulated an increasing number of effective B cells as more antigens were loaded by dendritic cells. In the single infection, the number of effective B cells rises from day 2 and peaks at day 8 according to antigens loaded by dendritic cells (figure 5*c*). However, as B cells produce effective Abs for clearing virus, it was expected to decrease after the peak amount of B cells since virus clearance reduces antigens loaded for B cell activation [14]. The peak B cells occurring in day 8 are consistent with reports that viral antigens are hardly detected beyond this day [32]. The modelling produced these results as the effective number of B cells gradually declined after day 8 until the end of the period of infection. Although repeated

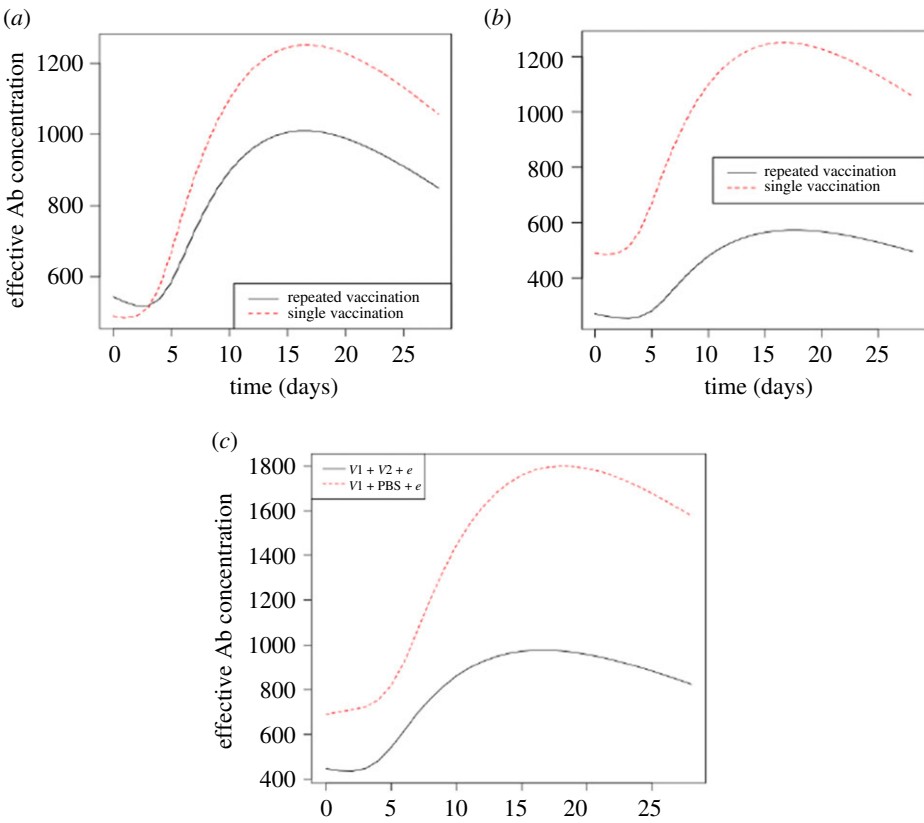

**Figure 6.** Antibodies produced in response to epidemic infection. (*a*) Effective antibodies (pg ml$^{-1}$) produced in repeated vaccination when prior vaccine strains cross-react at 10%. (*b*) Effective antibodies (pg ml$^{-1}$) produced in repeated vaccination when prior vaccine strains cross-react at 20%. (*c*) Effective antibodies (pg ml$^{-1}$) produced during epidemic following sequential infections. $V1 + V2 + e$ is repeated infection and $V1 + PBS + e$ is single infection.

infections followed a similar trend, excessive tregs activated during the epidemic infection decreased antigens loaded by dendritic cells to activate B cells. This accounted for the lower numbers of effective tregs observed in the repeated infection compared with the single infection during the epidemic period of infections (figure 5*c*). During the early days of epidemic strain infection, lower antigen dose was loaded by dendritic cells to activate specific lymphocytes leading to lower numbers of B cells in both vaccination and sequential infection experiments at that time (figure 5).

With regards to the production of Abs, the effective Abs produced by the B cells were expected to increase at the time when the B cells increase as the epidemic progresses [33]. Besides, they were expected to decline as B cells also declined after peak days of B cells. Consistent with these expectations, effective Abs increased over the period while B cells were increasing in each experiment (figure 6). More importantly, repeated vaccinations in which vaccine strains cross-react at 10% had comparatively higher effective Abs within the first 3 days of virus infection but fall below the quantities produced in single vaccination after this time (figure 6*a*). This, particularly, higher effective Abs at the early days in repeated infection results from sufficient antigen dose loading by dendritic cells for higher production of Abs in the repeated vaccination where vaccines cross-react at 10%. Nevertheless, the Abs in both repeated and single vaccinations increase to a maximum from day 3 to day 15 after which their levels decline as a result of virus clearance (figure 6). Furthermore, single vaccination had higher effective Abs over the period of epidemic infection compared with repeated vaccination, suggesting that they provided a better reaction to the infections. The study revealed that the second vaccination decreased the fraction of sites on the surfaces of dendritic cells that load antigens. This decreases the antigen dose loaded by dendritic cells to activate effective lymphocytes during the epidemic infection. In effect, relatively higher antigen dose loaded by dendritic cells in single infection resulted in activating higher numbers of effective lymphocytes compared with the repeated vaccination. This accounts for the activated effective tregs and B cells as well as Abs produced in single vaccinations. It is worth noting that in the experiment involving vaccine–vaccine cross-reactivity of 20%, although repeated vaccination had lower effective tregs, Abs produced were not sufficient to react to the epidemic strain permitting higher viral loads compared with

single infection (figure 3). Lund and colleagues reported a similar observation when depletion of tregs in a virus infection resulted in higher viral loads [34,35].

Similarly, results of effective Abs produced in repeated infections are lower compared with single infection (figure 6c). In this case, antigen presentation by dendritic cells is greatly suppressed by the higher numbers of effective tregs for production of Abs in repeated infection compared with single infection. The observation is well noted in the role of tregs to dynamically control the primary immune response [36–38]. As in the case following vaccination, there is low virus concentration at the early days of infection resulting in lower antigen dose loaded by dendritic cells. This results in the activation of the low B cells and consequently the production of Abs at the early days of infection in both repeated and single infection (figure 6c). Overall, lower levels of effective Abs were found in repeated infections compared with single infection as a result of relatively lower amounts of B cells. These suggest that Abs produced in response to the epidemic result in a better protection in single infection compared with repeated infection. In essence, the dynamics of Abs production make repeated infection more susceptible during the epidemic compared with single infection because the suboptimal reaction to the virus leads to the higher viral loads.

## 3.4. Virus loads and T-regulatory cells of cross-reacting infections

The sensitivity of viral loads to cross-reactive rates was examined by studying the effects of sequential infections of several cross-reactive strains on viral loads using the mathematical model. This was simulated in a sequential infection with $V1$ for 28 days followed by $V2$ for another 28 days. The host was then injected with the challenge virus, $e$ (epidemic strain). The experiments are repeated for 100 pairs of cross-reactive rates. Each pair of values of cross-reactivity represents the chances that lymphocytes derived by $V1$ will recognize $e$ ($V1$-$e$ cross-reactivity) and the chances that lymphocytes derived by $V2$ infection will recognize $e$ ($V2$-$e$ cross-reactivity), respectively. The peak virus load for each experiment and the number of effective tregs are then derived.

The viral loads decreased as the $V1$-$e$ cross-reactivity and $V2$-$e$ cross-reactivity increased (figure 7a). Changes in viral loads as the $V1$-$e$ cross-reactivity increases indicate that viral loads are very sensitive to the $V1$-$e$ cross-reactivity. Increasing the $V1$-$e$ cross-reactivity increased the effective Abs that reacted to the virus leading to the decrease in viral loads. On the other hand, while the $V2$-$e$ cross-reactivity was increased, viral loads decreased significantly for cross-reactivity exceeding 30% (figure 7a). The decrease in the viral loads was a result of increase in effective Abs that react to the virus as the $V2$-$e$ cross-reactivity increases.

In investigating the roles of tregs in relation to antigens and regulation of the immune response, tregs were expected to be sensitive to the cross-reactive rates between infecting strains [39,40]. This was observed as number of effective tregs changed with $V2$-$e$ cross-reactivity increased (figure 7b). The increase in the effective tregs as $V2$-$e$ cross-reactivity increases is achieved as a result of reactivating $V2$-specific tregs that complement the effective tregs derived during the epidemic. Nevertheless, while varying the $V1$-$e$ cross-reactivity, only small increments in tregs in some instances of $V2$-$e$ cross-reactivity were observed (figure 7b). The $V1$-derived tregs directly add to the effective tregs activated during the epidemic infection. Therefore higher $V2$-$e$ cross-reactive rate increases the effective tregs derived at the epidemic infection by reactivating more $V2$-specific tregs. However, $V2$-derived tregs occur in relatively larger amounts so that higher $V1$-$e$ cross-reactivity will be required to observe its accompanying changes during the epidemic. The consequences are that the changes in tregs that occurred due to increasing $V1$-$e$ cross-reactivity required higher $V2$-$e$ cross-reactivity. These results further confirm earlier observation that the immune response is sensitive to changes in $V1$-$e$ cross-reactivity as it does to $V2$-$e$ cross-reactivity.

## 3.5. Implications of findings on influenza vaccinations

Epidemiological findings following influenza vaccinations linked individuals who had been vaccinated with variable risks of influenza infection in an epidemic [4–7]. In this study, it is suggested that repeated vaccination increases susceptibility to infection at the instance of residual immunity induced by the vaccination.

In order to allow a direct comparison of the study with the work of Skowronski and colleagues [4], additional experiment was performed in which the first and second vaccine strains were identical, i.e. cross-reactive rate between the first and second vaccine strains is 100%. In this case, the viral loads were also greater in the repeated vaccinations (figure 8). This supports the earlier results of the investigations involving vaccine strains which cross-react at 20%. Additional supporting results on viral loads corresponding with other vaccine–vaccine cross-reactive rates are presented in electronic

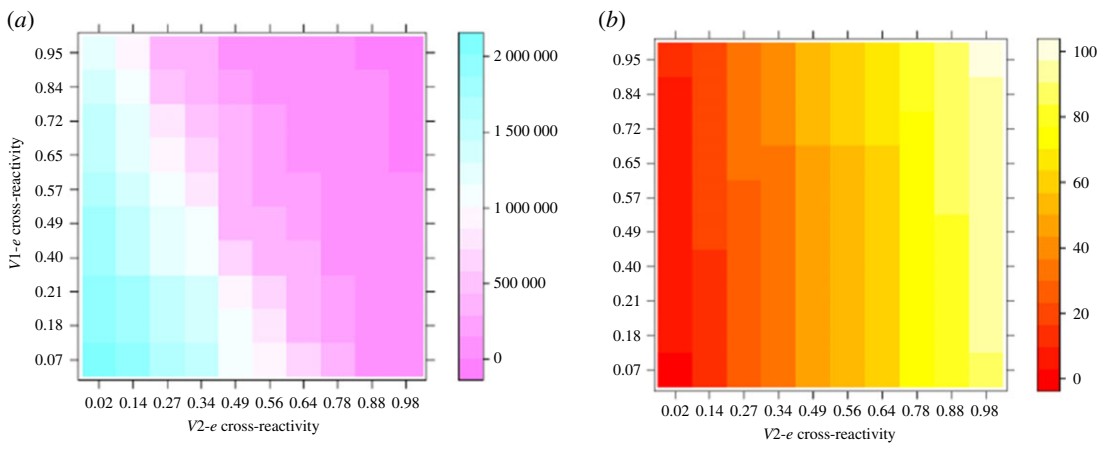

**Figure 7.** Sensitivities of viral loads (*a*) and T-regulatory cells (*b*) to cross-reactivity between epidemic and prior infection strains. (*a*) Viral loads are sensitive to cross-reactive rates of infections. (*b*) T-regulatory cells are sensitive to cross-reactive rates of infections. Results involve differing cross-reactive rates between a first infecting strain (*V1*) and the epidemic (challenge) strain *e* (*V1-e* cross-reactivity) and the cross-reactive rates between the second infecting strain (*V2*) and the epidemic strain *e* (*V2-e* cross-reactivity).

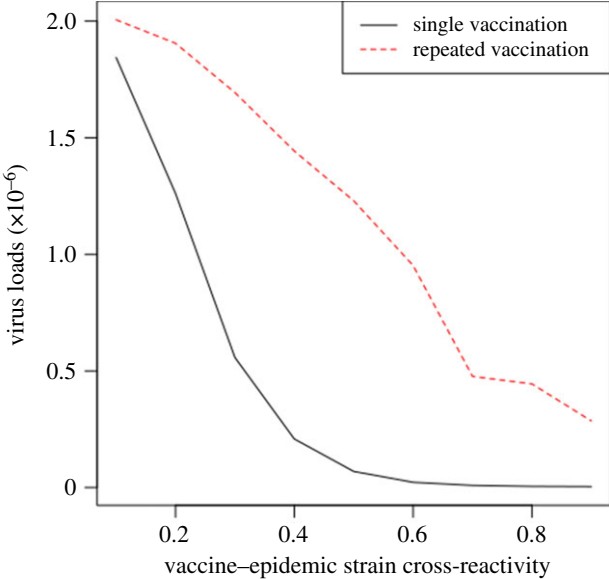

**Figure 8.** Repeated vaccination has higher viral loads ($EID_{50}$ $ml^{-1}$) compared with single vaccination. Vaccine strains cross-react at 100%.

supplementary material, figure S1. The presence of higher viral loads during the epidemic corresponds with the low vaccine effectiveness in repeated vaccination as in [4]. Particularly, using published data from an influenza vaccine effectiveness study in Canada [4], it was found that whereas the unadjusted vaccine effectiveness (VE) among those who received both 2013–2014 and 2014–2015 vaccine against influenza A (H3N2) is −0.33, VE among those who received only the 2013–2014 but not 2014–2015 seasons vaccine is −0.28. These results provide further support to the findings that single vaccination are better protected by vaccination compared with the repeated vaccination during the epidemic. Overall, the findings on viral loads suggest that the residual immunity conferred by repeated vaccination makes vaccine recipients more susceptible to infections during the influenza seasons.

## 4. Discussion

Sometimes, rather than boosting immunity, repeated vaccination make vaccine recipients more susceptible to epidemic infections compared with single vaccination. These have been reported in the case of seasonal influenza infestation over the past years [4–7]. These have also been observed during dengue virus infection [3] and malaria among others [41–43]. The mechanisms underlying the

defective immune response following the vaccinations are less understood [44]. This work investigated the dynamics of some components of the immune response that make recipients of repeated vaccination more susceptible in seasonal influenza epidemic infections.

The study identifies higher viral loads in repeated vaccinations compared with single vaccination as a factor contributing to the susceptibility of the vaccine recipients during influenza epidemics. In vaccination, the activated effective lymphocytes that react to the epidemic strain are reduced resulting in higher viral loads compared with sequential infections. The fraction of sites on the surfaces of dendritic cells that load antigens of the epidemic strains are reduced in the vaccinations. This lower antigen dose to be loaded by dendritic cells is to activate effective B cells and Abs. Insufficient effective Abs allow virus to replicate leading to high viral loads. Although repeated vaccinations could enhance the production of B cells and Abs, the cross-reactive rates between vaccine strains were very crucial during the epidemic infections. In particular, vaccine strains that cross-react at 20% culminate in decreasing antigens loaded by dendritic cells for activating the effective B cells and to produce Abs in the epidemic. Therefore, there are lower effective B cells and Abs during the epidemic compared with experiments in which vaccine strains cross-react at 10% (figures 5 and 6).

The study further investigated the immune response to repeated infections. Individuals who have experienced repeated prior infections are highly susceptible to infections during the epidemic compared with those with single prior infection, which have much lower viral loads. In repeated infections, a cross-reactive epidemic strain reactivates tregs derived from previous infections. This increases the effective tregs activated in response to the epidemic in repeated infections. Also, increasing the prior–epidemic strain cross-reactivity increases the number of reactivated tregs which cumulates the effective tregs derived during the epidemic infection. Therefore, the effective number of tregs during the epidemic increases as the cross-reactivity between the prior infections and the epidemic strain increases (figure 4). A second infection that cross-reacts with the first infection at a rate of 20% activates an excessive number of tregs compared with a second infection which cross-reacts at a rate of 10% (figure 4).

Consequently, during the epidemic infection, numbers of effective tregs are produced in excess in the repeated infections involving prior infecting strains that cross-react at 20% compared with 10% cross-reactive rate. The higher effective tregs decrease the antigen dose of epidemic strain loaded by dendritic cells to activate lymphocytes. This leads to lowering the activated number of effective B cells and the production of Abs during the epidemic infections. As the epidemic progresses, higher antigen dose are loaded by dendritic cells in both groups (single and repeated infections) resulting in the activation of higher B cells and Abs production until reduction in viral loads as Abs react to the virus (figures 5 and 6). Both B cells and Abs produced are decreased since antigen dose decreases because of virus clearance (in single and repeated infections). However, sufficient effective tregs in single infections permit the highest antigen dose to be loaded by dendritic cells to activate higher B cells to produce Abs compared with repeated infections. Such role of tregs in affecting antigen presentation in the process of immune response is also well reported in [45–48].

The presence of higher Abs in single infections provides better reaction to the epidemic strain resulting in lowering viral loads. Specifically, 20% cross-reactivity between the prior infections and the epidemic strain is sufficient to increase the effective B cells to produce Abs capable of completely clearing the epidemic strains in the single infection. The effective Abs are increased by increasing the cross-reactivity between the prior infections and the epidemic strain to 30% or more for a complete virus clearance in repeated infections. In addition, when prior infections cross-react with the epidemic strain at a rate less than 6%, effective lymphocytes activated in the early days of repeated infections are in very low amounts. This allowed the repeated infections to have more than twice the viral loads in single infections as virus replicates (figure 3d). However, this is decreased as the cross-reactivity between the prior infections and the epidemic strain increases since these increments allow more Abs to be produced to react to the virus. The presence of higher viral loads in the repeated infection for a very low prior–epidemic strain cross-reactivity make them more susceptible to infection compared with single infection. On the detrimental effect of excess suppression of antigen dose loading by tregs, it has been suggested that increasing antigen dose can diminish the effect of tregs in order to allow for activation of sufficient lymphocytes in response to antigens [23]. In this regard, vaccines with adjuvants intended to boost loading of antigen dose by dendritic cells can be used to reduce enhancements of infection through residual tregs during the epidemic.

The findings of this study suggest that repeated vaccination could enhance immune response provided that the vaccine strains cross-react at rates which do not exceed 10%. Nonetheless, as antigen dose of dendritic cells increases as the epidemic progresses with virus replication, more Abs are produced (in single vaccination) to react to the virus (compared with the repeated vaccination). The presence of higher

effective B cells and effective Abs produced as the epidemic progresses beyond day 3 provide better protection for individuals with single vaccination compared with those who receive repeated vaccinations. These results support the previous studies that found that repeated vaccination was not effective in the long term [9,49]. In providing insights into these conditions in vaccination, this study has analysed the effect of cross-reactive strains on the human humoral aspects of immune response. However, this contrasts another study of the effectiveness of repeated influenza vaccination in which antigenic distance between strains was suggested to cause a variable efficacy in repeated vaccination using a computer model [29]. Furthermore, using theoretical models, this study analysed roles of tregs suppression in the investigation of immune response to repeated vaccinations and infections. Nevertheless, it may be possible to consider other models that accentuate antibody affinity for such study as has been done in original antigenic sin [50].

## 5. Conclusion

Overall, the study suggests probable characteristics of vaccine or infection strains (viruses) that make repeated vaccination/infection detrimental. In particular, the results of the study suggest that individuals with single vaccination are less susceptible to infections during the epidemic compared with those who receive repeated vaccination with residual immunity gained from first vaccination. It has been shown that the higher effective Abs produced to react to the epidemic infection make the immune response more effective in single prior vaccination and infection. The integration of literature and simulations with the models are indicative that the immune response may be compromised in repeated vaccination during the influenza epidemic compared with those with lesser vaccination. In repeated vaccinations, although immunity may be enhanced at the early days of the epidemic with vaccine strains that cross-react at 10% or less, those with single vaccination will always have better protection in the long term. The findings of this study call for further investigation and considerations in vaccine designs and administration. Particularly, since antigen dose loading by dendritic cells are reduced in repeated vaccination, this can be heightened by using adjuvanted vaccines so as to improve the activated effective lymphocytes during the epidemic. This could have been applied during the 2014–2015 influenza season where most of the vaccines were not adjuvanted. This study investigated the dynamics of the immune response to influenza infection and use can be made of the methods in studying other variable pathogens.

Data accessibility. The data and sources of data used in the paper are included in this submission and other supplementary results are uploaded as part of the electronic supplementary material.

Competing interests. There are no competing financial or other interests in relation to this work.

Funding. This work was supported by a grant from the International Development Research Centre (IDRC) to the African Institute for Mathematical Sciences Next Einstein Initiative (AIMS NEI).

Acknowledgement. The author thanks Wilfred Ndifon for his constructive comments on an earlier version of the paper.

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
