## [Peer Review File · Royal Society Open Science]

Review History

RSOS-201433.R0 (Original submission)

Review form: Reviewer 1

Is the manuscript scientifically sound in its present form?

No

Are the interpretations and conclusions justified by the results?

No

Is the language acceptable?

Yes

Do you have any ethical concerns with this paper?

No

Have you any concerns about statistical analyses in this paper?

Yes

Recommendation?

Reject

Comments to the Author(s)

This is an very interesting topic. While I definitely endorse the importance of the research question and the modeling approach adopted by the author, I am not convinced by this work for two reasons:

1. the model structure does not sufficiently account for repeated infections and vaccinations. For instance, innate, adaptive and memory immunity as well as their interactions was not adequately considered and modeled after each infection and vaccination. How about the time-evolving molecular level immunity profile (e.g., epitope)? There are many more questions I can go on and on, but I will just give some examples here.
2. No data was used to validate the modeling results and it's just hard for me to believe the results in their current form.

Review form: Reviewer 2**Is the manuscript scientifically sound in its present form?**

Yes

Are the interpretations and conclusions justified by the results?

Yes

Is the language acceptable?

Yes

Do you have any ethical concerns with this paper?

No

Have you any concerns about statistical analyses in this paper?

Yes

Recommendation?

Accept with minor revision (please list in comments)

Comments to the Author(s)

The manuscript is well written and addresses a very interesting, fundamental, novel and topically subject concerning the phenomenon whereby some infections and vaccines can induced reduced protection to subsequent infection. The paper is focused on influenza virus infection/vaccines and uses mathematically models to assess the response to vaccine scenarios and explain why vaccines may be less efficacious on related use. The mathematics are beyond me but I found the paper intriguing. Being unfamiliar with these models I would like to understand why the authors assume Tregs are the main cause on immune suppression in this model and why the models don't address antibody binding affinity with the developing immune response.

It got to be accepted that the results of these studies, while thoughtful and though provoking, are nevertheless theoretical and not proven by experiment, Thus I believe the authors should tone down their final conclusion such as

'Overall, the study has elucidated the precise characteristics of vaccine or infection strains (viruses) that make repeated vaccination/infection detrimental'

Then study suggests rather than elucidates putative means by which repeated vaccine may cause failure of repeat vaccination.

Decision letter (RSOS-201433.R0)

Dear Dr Adabor

On behalf of the Editors, we are pleased to inform you that your Manuscript RSOS-201433 "Computational investigations of the immune response to repeated influenza infections and vaccinations" has been accepted for publication in Royal Society Open Science subject to minor revision in accordance with the referees' reports. Please find the referees' comments along with any feedback from the Editors below my signature.

Please submit your revised manuscript and required files (see below) no later than 7 days from today's (ie 04-Dec-2020) date. Note: the ScholarOne system will 'lock' if submission of the revision is attempted 7 or more days after the deadline. If you do not think you will be able to meet this deadline please contact the editorial office immediately.

on behalf of Professor Tim Rogers (Associate Editor) and Mark Chaplain (Subject Editor)
openscience@royalsociety.org

Subject Editor (Professor Mark Chaplain):

Comments to the Author:

Neither referee has any issue with the mathematical content, but both are concerned about the extent to which the model is realistic and can be used to draw conclusions about real-world infections.

Before the paper can be accepted for publication, it must be modified to include a detailed discussion of these issues - in particular in response to the concerns of referee 1 about the details that have been left out of the model.

Thank you.

Reviewer comments to Author:

Reviewer: 1

Comments to the Author(s)

This is a very interesting topic. While I definitely endorse the importance of the research question and the modeling approach adopted by the author, I am not convinced by this work for two reasons:

1. the model structure does not sufficiently account for repeated infections and vaccinations. For instance, innate, adaptive and memory immunity as well as their interactions was not adequately considered and modeled after each infection and vaccination. How about the time-evolving molecular level immunity profile (e.g., epitope)? There are many more questions I can go on and on, but I will just give some examples here.
2. No data was used to validate the modeling results and it's just hard for me to believe the results in their current form.

Reviewer: 2

Comments to the Author(s)

The manuscript is well written and addresses a very interesting, fundamental, novel and topically subject concerning the phenomenon whereby some infections and vaccines can induce reduced protection to subsequent infection. The paper is focused on influenza virus infection/vaccines and uses mathematical models to assess the response to vaccine scenarios and explain why vaccines may be less efficacious on related use. The mathematics are beyond me but I found the paper intriguing. Being unfamiliar with these models I would like to understand why the authors assume Tregs are the main cause on immune suppression in this model and why the models don't address antibody binding affinity with the developing immune response.

It got to be accepted that the results of these studies, while thoughtful and though provoking, are nevertheless theoretical and not proven by experiment, Thus I believe the authors should tone down their final conclusion such as

'Overall, the study has elucidated the precise characteristics of vaccine or infection strains (viruses) that make repeated vaccination/infection detrimental'

Then study suggests rather than elucidates putative means by which repeated vaccine may cause failure of repeat vaccination.

===PREPARING YOUR MANUSCRIPT===

===PREPARING YOUR REVISION IN SCHOLARONE===

- Any electronic supplementary material (ESM).
- If you are requesting a discretionary waiver for the article processing charge, the waiver form must be included at this step.
- If you are providing image files for potential cover images, please upload these at this step, and inform the editorial office you have done so. You must hold the copyright to any image provided.
- A copy of your point-by-point response to referees and Editors. This will expedite the preparation of your proof.

- Ensure that your data access statement meets the requirements at <https://royalsociety.org/journals/authors/author-guidelines/#data>. You should ensure that you cite the dataset in your reference list. If you have deposited data etc in the Dryad repository, please only include the 'For publication' link at this stage. You should remove the 'For review' link.
- If you are requesting an article processing charge waiver, you must select the relevant waiver option (if requesting a discretionary waiver, the form should have been uploaded at Step 3 'File upload' above).
- If you have uploaded ESM files, please ensure you follow the guidance at <https://royalsociety.org/journals/authors/author-guidelines/#supplementary-material> to include a suitable title and informative caption. An example of appropriate titling and captioning may be found at https://figshare.com/articles/Table_S2_from_Is_there_a_trade-off_between_peak_performance_and_performance_breadth_across_temperatures_for_aerobic_scope_in_teleost_fishes_/3843624.

Author's Response to Decision Letter for (RSOS-201433.R0)

See Appendix A.

RSOS-201433.R1 (Revision)

Review form: Reviewer 2

Is the manuscript scientifically sound in its present form?

Yes

Are the interpretations and conclusions justified by the results?

Yes

Is the language acceptable?

Yes

Do you have any ethical concerns with this paper?

No

Have you any concerns about statistical analyses in this paper?

No

Recommendation?

Accept as is

Comments to the Author(s)

The authors have addressed my concerns raised in my previous review.

Decision letter (RSOS-201433.R1)

Dear Dr Adabor,

It is a pleasure to accept your manuscript entitled "Computational investigations of the immune response to repeated influenza infections and vaccinations" in its current form for publication in Royal Society Open Science. The comments of the reviewer(s) who reviewed your manuscript are included at the foot of this letter.

You can expect to receive a proof of your article in the near future. Please contact the editorial office (openscience@royalsociety.org) and the production office (openscience_proofs@royalsociety.org) to let us know if you are likely to be away from e-mail contact – if you are going to be away, please nominate a co-author (if available) to manage the proofing process, and ensure they are copied into your email to the journal.

Kind regards,
Anita Kristiansen

Editorial Coordinator

on behalf of Professor Tim Rogers (Associate Editor) and Mark Chaplain (Subject Editor)
openscience@royalsociety.org

Reviewer comments to Author:
Reviewer: 2

Comments to the Author(s)
The authors have addressed my concerns raised in my previous review.

Appendix A

RESPONSE TO REFEREES

Subject Editor:

Neither referee has any issue with the mathematical content, but both are concerned about the extent to which the model is realistic and can be used to draw conclusions about real-world infections.

Before the paper can be accepted for publication, it must be modified to include a detailed discussion of these issues - in particular in response to the concerns of referee 1 about the details that have been left out of the model.

Response to subject editor: In the revised manuscript, concerns raised by the reviewers have been thoroughly addressed. Appropriate responses and changes to the manuscript are provided to each comment of the reviewers.

Reviewer #1:

(1) The model structure does not sufficiently account for repeated infections and vaccinations. For instance, innate, adaptive and memory immunity as well as their interactions was not adequately considered and modeled after each infection and vaccination. How about the time-evolving molecular level immunity profile (e.g., epitope)? There are many more questions I can go on and on, but I will just give some examples here.

Response: In line with previous studies [refs 18-20], this study focuses on humoral aspects of adaptive immunity. For a strain V1, if the adaptive immune system has never encountered any antigen which cross-reacts with V1, specific naive lymphocytes will be activated [refs 21,22]. When a strain V2, which cross-reacts with V1, is encountered by the immune system in future infection, then a subset of V1-specific lymphocytes, including tregs will be re-activated in addition to the activation naïve lymphocytes by V2. The reactivated Tregs will suppress the presentation of V2-specific antigens by dendritic cells and decrease antigen dose loading of V2 accessible for activating V2-specific B cells and specific T-helper cells [refs 22,23]. If a strain V3 that cross-reacts with V1 and V2 is encountered, then the strength of humoral response to V3 will depend on activated V1- and V2- specific B cells and naive lymphocytes elicited by V3 [ref 23]. Note that, a much lower antigen dose will be needed to reactivate the V1- and V2- specific lymphocytes compared to naive lymphocytes [ref 24]. These facts about humoral aspects of adaptive immunity form the structure of the model. In particular, it covers interactions between the virus, its target cells and the regulation of the humoral response by tregs. Therefore, the equations of the simplified model (Equations (1)-(9)) were formulated based on these interactions. This approach is consistent with other prior studies and sufficient in details regarding the scope of this study [refs 18-20]. In the model, cross-reactive rates specify cross-reactivity between strains. Here, two strains cross-react when lymphocytes activated by one strain can recognize the other strain. These additional insights about the model have been added to Section 2.1 (pages 4, 5 and 6) of the revised manuscript. Further details about the model simulation of sequential infection and vaccination are presented in Section 2.2.

Additionally, the modeling account for interactions described above in repeated infection/vaccination in that each infection/vaccination is distinguished by a subscript ($i/j=1,2,3$). For instance, if you consider Equation 5, there will be a rate of change of B cells for each subscript, i . The structure of the model presentation is sufficient and consistent with previous simulation studies [ref 25].

(2) No data was used to validate the modeling results and it's just hard for me to believe the results in their current form.

Response: Additional results (Figure 2) have been provided to show that the model reasonably fit experimental data that describe influenza virus and IgG antibody concentrations measured in the lungs and sera of infected mice reported in previous studies [data available in ref 26]. Details of the new results are available on page 8 of the revised manuscript. The use of these results to support such modeling studies is consistent with literature or studies conducted in the past (See refs 25 and 26). Furthermore, sufficient references have been provided to support model relationships and results. These are in line with literature and gives confidence in the results of the study.

Reviewer #2:

1. Being unfamiliar with these models I would like to understand why the authors assume Tregs are the main cause on immune suppression in this model and why the models don't address antibody binding affinity with the developing immune response.

Response: The assumption of tregs-mediated suppression was informed by its role in suppressing the immune response using a variety of mechanisms reported in previous studies [refs 36, 37 and 22]. Please kindly refer to the role of tregs applied to this study as part of first response to Reviewer #1. This is detailed on page 4 of the revised manuscript. The study focused on the role of tregs suppression in the investigation on immune response to repeated vaccinations and infections. Therefore, the theoretical model and analysis were designed along this line of study. Nevertheless, it may be possible to consider other models that accentuate antibody affinity for such study as has been done in antigenic sin [ref 50]. These distinctions are presented in page 28 of revised manuscript.

2. It got to be accepted that the results of these studies, while thoughtful and though provoking, are nevertheless theoretical and not proven by experiment, Thus I believe the authors should tone down their final conclusion such as

'Overall, the study has elucidated the precise characteristics of vaccine or infection strains (viruses) that make repeated vaccination/infection detrimental'

Then study suggests rather than elucidates putative means by which repeated vaccine may cause failure of repeat vaccination.

Response: Appropriate words have been used in the conclusion of the revised paper in response to this suggestion from the reviewer. In particular, the following sentence "Overall, the study has elucidated the precise characteristics of vaccine or infection strains (viruses) that make repeated vaccination/infection detrimental" has been replaced with "Overall, the study suggests probable characteristics of vaccine or infection strains (viruses) that make repeated vaccination/infection detrimental." Other parts of the manuscript that needed such revisions have been corrected in the revised paper.